# Anomalous Concentration Dependence of Surface Tension and Concentration-Concentration Correlation Functions of Binary Non-Electrolyte Solutions

**DOI:** 10.3390/ijms24032276

**Published:** 2023-01-23

**Authors:** Carlo Carbone, Eduardo Guzmán, Ramón G. Rubio

**Affiliations:** 1Departamento de Química Física, Facultad de Ciencias Químicas, Universidad Complutense de Madrid, Ciudad Universitaria s/n, 28040 Madrid, Spain; 2Instituto Pluridisciplinar, Universidad Complutense de Madrid, Paseo de Juan XXIII 1, 28040 Madrid, Spain

**Keywords:** binary mixtures, critical point, fluctuations, non-electrolytes, surface tension

## Abstract

The concentration dependence of the surface tension of several binary mixtures of non-electrolytes has been measured at 298.15 K. The mixtures have been chosen since they presented a so-called “W-shape” concentration dependence of the excess constant pressure heat capacity and high values of the concentration-concentration correlation function. This behavior was interpreted in terms of the existence of anomalously high concentration fluctuations that resemble those existing in the proximities of critical points. However, no liquid-liquid phase separation has been found in any of these mixtures over a wide temperature range. In this work, we have extended these studies to the liquid-air interfacial properties. The results show that the concentration dependence of the surface tension shows a plateau and the mixing surface tension presents a “W-shape” behavior. To the best of our knowledge, this is the first time that this behavior is reported. The weak anomalies of the surface tension near a liquid-liquid critical point suggest that the results obtained cannot be considered far-from-critical effects. The usual approach of substituting the activity by the concentration in the Gibbs equation for the relative surface concentration has been found to lead to large errors and the mixtures to have a fuzzy and thick liquid/vapor interface.

## 1. Introduction

In the last few years, several works showed that the excess constant pressure heat capacity, $C_{p}^{E},$ of some binary mixtures of non-electrolytes presented an anomalous concentration dependence [1,2,3]. More specifically, this dependence had a more or less pronounced W-shape. The molecules of the studied mixtures did not have any chemical structural correlation. A very extensive description of the fluctuation theory of mixtures and its consequences on the thermodynamic properties of liquid mixtures is provided by Matteoli and Mansoori [4]. Depolarized light scattering experiments performed by Rubio et al. [5] allowed for the determination of the so-called concentration-concentration correlation function, S_cc_, which is the zero-wave vector limit of the structure factor. The authors pointed out that, for these binary mixtures, S_cc_ took values significantly higher than those corresponding to an ideal mixture, thus indicating that the concentration fluctuations were high. Therefore, at the molecular level, the composition was clearly non-random, thus differing from the overall one. Experimental evidence of the difference between bulk and local concentrations has been provided by different authors using spectroscopic methods [6,7,8], as well as computer simulations [9,10,11] and theoretical calculations [12,13,14,15]. Notably, S_cc_ is the inverse darken stability [16,17,18,19] and is closely related to the osmotic compressibility, ${\left(\frac{\partial \mu_{i}}{\partial x_{i}}\right)}_{T,P}$, with μ_i_ being the chemical potential. It is well known that S_cc_(0) values much higher than the one for ideal mixtures (max. S_cc_(0) = 0.25 for x = 0.5) are typical of the systems near a critical point. However, the decrease in the temperature of the studied mixtures did not lead to any phase separation; therefore, far-from-critical effects are difficult to be considered for justifying the results found. 

It is well known that in binary mixtures the osmotic compressibility gives the excess surface concentration of component i at the vapor/liquid interface. In most cases reported in the literature, an ideal behavior is defined as $\mu_{i}\left(T,P\right)=\mu_{i}^{0}\left(T,P=1\mathrm{bar}\right)+\mathrm{RT}\ln x_{i}$, with x_i_ being the mole fraction of component i. However, Coto et al. [20] have shown that assuming an ideal behavior to calculate the osmotic compressibility may be inadequate for calculating surface relative adsorption, Γ_2,1_, which is also observed for dilute water-alcohol mixtures. Using small angle neutron scattering, Almasi et al. [21] have shown that small errors in the excess Gibbs energy, G^E^, may lead to large errors in ${\left(\frac{\partial \mu_{2}}{\partial x_{2}}\right)}_{T}$. Ritacco et al. [22,23] reported that aqueous trisiloxane solutions are an example of this problem from ellipsometry measurements. Similar conclusions were obtained from Llamas et al. [24,25] by combining surface tension and neutron reflectometry measurements.

Concentration fluctuations have been found to have strong consequences on other thermodynamic properties, even relatively far from any critical point, and for mixtures in which strong interactions, e.g., hydrogen bonds, exist. An example is the mixture that shows W-shape $C_{p}^{E}$ curves. Similar conclusions have been reached for aqueous solutions [26,27,28,29,30,31,32,33,34].

The adsorbed amount of component 2 relative to component 1, Γ_2,1_, is directly related to the composition dependence of the surface tension and chemical potential, μ_2_, through the Gibbs equation:(1)$$\Gamma_{2,1}=-{\left(\frac{\partial \gamma }{\partial \mu_{2}}\right)}_{T}.$$

The difficulty in precisely calculating the chemical potential from experimental data, has most frequently led researchers to assume an ideal behavior for the mixtures, thus Equation (1) can be rewritten as [35]:(2)$$\Gamma_{2,1}=-\frac{1}{\mathrm{RT}}x_{2}{\left(\frac{\partial \gamma }{\partial x_{2}}\right)}_{T}.$$

In general, it is clear that Equation (2) cannot be applied to concentrated mixtures, such as those studied in this work. Despite some approximate methods that have been proposed for calculating μ_2_ [36], we have preferred to calculate Γ_2,1_ using experimental data, thus not relying on any theoretical approximation.

More direct methods for measuring Γ_2,1_ have been described in the literature. Ellipsometry [37,38] and neutron or X-ray reflectometry [39,40,41,42,43], as well as radiative decomposition of tritium-doped molecules [44] have provided reliable values of Γ_2,1_ for binary mixtures, including dilute solutions of soluble surfactants. However, in some cases, e.g., for ellipsometry or reflectometry, it is necessary to assume the validity of a structural model for the interface, and then to fit the experimental data to the predictions of the model. Moreover, in the case of ellipsometry, the small difference in the refractive indexes of the components in each mixture does not allow one to obtain reliable values of the surface thickness.

Concentration fluctuations, as measured by the concentration-concentration correlation functions (zero wave-vector structure factor), are directly related to the osmotic compressibility by
(3)$$S_{\mathrm{cc}}^{-1}\left(0\right)=\frac{1}{{\mathrm{RTx}}_{1}x_{2}}{\left(\frac{\partial \mu_{2}}{\partial \ln x_{2}}\right)}_{T}$$

A brief description of the relationship of S_cc_(0) with the integral theory of fluids is provided in Appendix A.

It is well known that $S_{\mathrm{cc}}\left(0\right)$ diverges close to the critical points, whereas it takes a simple expression for ideal solutions, S_cc_(0) = x_1_·x_2_. The values of S_cc_(0) for the studied systems have been obtained from depolarized light scattering experiments, as described in detail in Appendix.

The overall goal of this manuscript is to study whether the systems that presented W-shape $C_{p}^{E}$ curves also show an anomalous behavior on the concentration dependence of the surface tension. The reason for this possibility is that, in equilibrium, μ_i_ (bulk) = μ_i_ (interface); therefore, strong concentration fluctuations in the bulk might affect the surface properties. Furthermore, we will discuss the excess consequences for the relative surface concentration behavior.

## 2. Materials and Methods

All solvents used in this work were purchased from Sigma-Aldrich (Saint Louis, MO, USA) and were of the maximum purity available, always exceeding 99.8 wt%. However, in the case of chloronaphtalene, the commercial purity was below 99 wt% in order that it was purified by five melting-thawing cycles until no impurities were detected by gas chromatography. Moreover, constant surface tension and density were observed after two consecutive cycles.

The surface tension was measured using three different tensiometers: Pendant drop, Wilhelmy plate, and de Nouy ring. The first was home made and the ADSA analysis was used for the analysis of the drop profile, as in previous works [24,25]. The Wilhelmy plate measurements were performed using a Nima model 702 Langmuir balance (Biolin, Göteborg, Sweden) and the ring experiments were carried out in a Krüs K-10 tensiometer (Krüs GmbH, Hamburg, Germany). Since the presence of impurities may lead to a time dependent surface tension due to the adsorption of the impurities, we have performed adsorption kinetic measurements for three mole fractions, 0.2, 0.5, and 0.8, for 2-butanone + n-decane, as well as chloronaphtalene + 2,2,4-trimethylpentane. The equilibration time in the plate and ring tensiometers were less than 2 min when each component has been thermostated prior to mixing. This is in accordance with the low viscosity and high diffusion coefficients of the solvents. Our techniques do not allow us to follow a sufficiently detailed kinetic measurement for these short stabilization times. Moving barrier measurements using a Langmuir balance induced a 400% reduction in the initial area. No change in the surface tension was detected within the experimental uncertainty. Therefore, we can conclude that the compounds were sufficiently pure for the surface tension experiments. 

The density measurements were carried out using an Anton Paar vibrating tube tensiometer DMA 4200-M (Anton Paar, Graz, Austria) with a precision of 3 × 10^−5^ g/cm^−3^.

The light scattering experiments were performed using an ALV/CGS-03 precision photogoniometer that uses a ALV-5000 correlator and a Glass-Thompson polarizer (ALV-Laser Vertriebsgesellschaft GmbH, Langen, Germany). This equipment was used in previous works [45] and includes a Coherent laser beam (Coherent Inc., Santa Clara, CA, USA) working at 532 nm. The refractive index was measured using a Carl-Zeiss refractometer (Jena, Germany) at 546 nm, with a precision of 10^−5^ units. Since it is important to obtain reliable values of S_cc_(0), independent values of $\left(\frac{\partial n}{\partial x_{2}}\right)$ were obtained using a BI-DNDC differential refractometer (Brookhaven Instrument Corporation, Holtsville, NY, USA) with a precision of 10^−3^ units.

Milli-Q water was used for cleansing with a resistivity higher than 18 MΩ·cm and an organic material content lower than 6 ppm. The glassware was cleaned with piranha solution (caution: Piranha solutions are dangerous), then thoroughly rinsed with pure water and dried in a vacuum oven.

## 3. Results and Discussion

Figure 1 shows the mole fraction dependence of the surface tension for the different studied mixtures. It is important to note the existence of a more or less extended pseudoplateau, which to date, has not been described in the literature. Even for mixtures close to their lower critical solution temperature (LCST), the pseudoplateau is not more pronounced [46]. Nevertheless, it must be noted that the experimental γ vs. X data do not show any maximum, minimum, or inflection point where ${\left(\frac{\partial \gamma }{\partial x}\right)}_{T,P}=0$, thus leading to $\Gamma_{2,1}=0$. This situation is found, e.g., in the case of aneotropic points, and has been discussed in detail in a book by Lyklema on binary mixtures of simpler compounds [47]. For completeness, we have included the results for the system methanol + methy-tertbutyl-ether (MTBE), in which no pseudoplateau is observed. Figure 1d shows that for the mixtures containing 1,2-dibromoethane, the plateau shifts to lower values of x_1_ as the chain length of the hydrocarbon decreases. On the other hand, the decrease in T shifts the pseudoplateau toward higher values of x1. Furthermore, as an example, Figure 1e shows that by lowering the temperatures of the mixtures no phase separation was found, as for any of the other mixtures. It has not been possible to describe the behavior observed using any of the available theoretical models within the experimental precision. The shape of the isotherms is similar to the type 2 gas adsorption isotherms (BET model) if γ-γ_2_ is plotted. Using this model as an empirical equation no good fit was obtained, even by using a maximum likelihood algorithm that includes the uncertainty in both dependent and independent variables [48]. Therefore, a pure empirical fitting approach has been used, while taking care that no numerical artifacts were introduced in the calculation of the derivative of γ vs. X.

In order to calculate ${\left(\frac{\partial \gamma }{\partial x_{2}}\right)}_{T,P}$ with sufficiently high precision, both the surface tension and the mixing surface tension, $\Delta \gamma =\gamma -X_{1}\cdot \gamma_{1}-X_{2}\cdot \gamma_{2}$, have been fitted to Padé approximants:(4)$$Y=\Lambda \frac{\sum_{i=0}^{n}A_{i}x_{1}^{i}}{1+\sum_{j=1}^{m}B_{j}x_{1}^{j}},.$$
where A_i_ and B_i_ are fitting parameters, Λ = 1 for the surface tension, γ, and Λ = x_1 × 2_ for the mixing surface tension, Δγ. Maximum values of 3 for n and m have been used for the sum of Equation (4), while ensuring that the uncertainties of the parameters used were less than 5%. For example, Figure 1a,b shows the experimental results and the corresponding fitted curves for the 1,4-dichlorobutane + n-heptane system, in which the plateau is more visible and has been experimentally mapped in more detail. It is clear that the fits are very good, and lead to randomly distributed residuals (not shown). For completeness, it is convenient to indicate that the experimental uncertainty on S_cc_(0) is about 3%, although it depends slightly on the difference in the refractive index of the compounds and their polarizability tensors.

As explained in Appendix A:(5)$$S_{\mathrm{cc}}^{-1}\left(0\right)=\frac{1}{{\mathrm{RTx}}_{1}x_{2}}{\left(\frac{\partial \mu_{2}}{\partial \ln x_{2}}\right)}_{T},$$
thus leading to
(6)$$\Gamma_{2,1}\left(x_{2}\right)=-x_{1}\frac{{\left(\frac{\partial \gamma }{\partial x_{2}}\right)}_{T,P}}{S_{\mathrm{cc}}\left(x_{2}\right)}.$$

Figure 2 shows the composition dependence of S_cc_(0) for the different studied mixtures. Some of the results were already published in a previous paper [5]; however, in order to have a more precise value for Γ_2,1_, we have performed additional experiments for better interpolation at the concentrations measured for γ. In all cases, it can be observed that the values are noticeably higher than the ideal mixture. Indeed, in some cases, the maxima of the curves are one order of magnitude higher, which as discussed below, has drastic consequences on the excess surface calculations.

Combining the values of ${\left(\frac{\partial \gamma }{\partial x_{2}}\right)}_{T,P}$ calculated from the experimental results and Equation (6), the excess surface concentration, Γ_2,1_, can be calculated from the experimental results. Notably, since the ${\left(\frac{\partial \gamma }{\partial x_{2}}\right)}_{T,P}$ is an experimental result independent of whether the mixture has been considered ideal or real, the ratio $\frac{\Gamma_{2,1}^{\mathrm{real}}\left(x_{2}\right)}{\Gamma_{2,1}^{\mathrm{ideal}}\left(x_{2}\right)}=\frac{S_{\mathrm{cc}}^{\mathrm{ideal}}\left(x_{2}\right)}{S_{\mathrm{cc}}^{\mathrm{real}}\left(x_{2}\right)}$ will give a direct information of how far the ideal mixture assumption is reliable. Since S_cc_(0) = x_1_·x_2_ for an ideal mixture, the experimental values shown in Figure 2 clearly point out that for the studied systems in this work, this approximation is not acceptable. An immediate conclusion is that the excess surface adsorption for these mixtures is smaller than the ideal mixtures except for the nitrobenzene + n-heptane in the high x_1_ region, as shown in Figure 3. A possible explanation is that the high values of the concentration-concentration correlation function indicate that the concentration which is very close to a molecule of type 2, X_2,local_, is quite different from the average macroscopic value, X_2_. However, X_2,local_ around one molecule of type 2 is not exactly the same as around another molecule of the same type at a different position at the same time. Ultimately, of course, the average value of X_2,local_ for all the molecules of type 2 has to be equal to the average value of X_2_.

## 4. Conclusions

The surface tension and the concentration-concentration has been measured for non-electrolyte binary mixtures that presented a “W-shape” $C_{p}^{E}$-composition dependence. An anomalous concentration dependence has been found both for the surface tension and the mixing surface tension. The first shows a plateau at intermediate compositions, and the mixing surface tension shows a marked shoulder in the plateau region. The concentration fluctuations have been obtained from depolarized light scattering experiments. The values obtained are well above the values corresponding to the ideal mixture. The combination of the two types of results clearly shows that the most usual approximation for calculating the relative excess surface concentration, in which the activities of the two components are substituted by mole fractions, is not acceptable. Indeed, the ratio between the values of the excess concentration calculated using the ideal mixture approximation and those rigorously calculated can be up to two orders of magnitude for the studied mixtures. Considering the high values of the concentration-concentration correlation function for the mixtures, associated with strong concentration fluctuations, these results can be associated with a high thickness of the liquid/vapor interface.

## Figures and Tables

**Figure 1 ijms-24-02276-f001:**
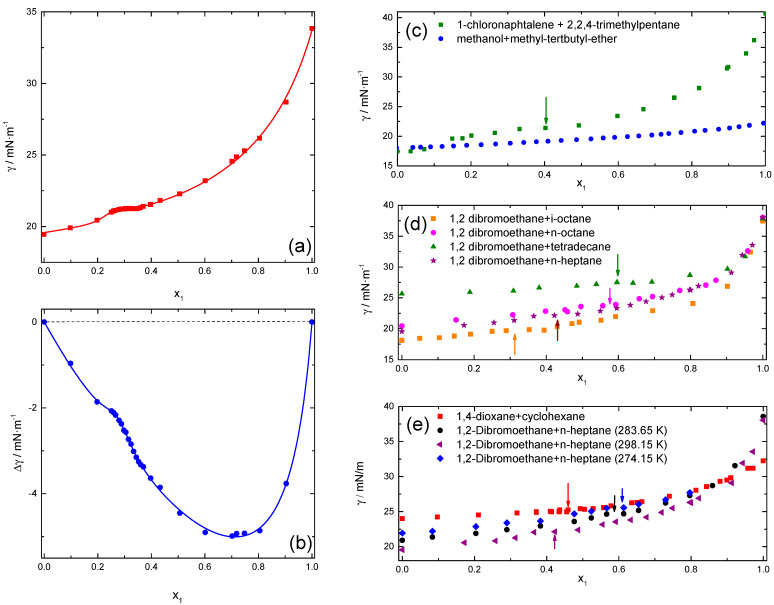
Composition of dependences of the surface tension (**a**) and mixing surface tension, Δγ, (**b**) at 298.15 K for 1,4-dichlorobutane + n-heptane mixtures. Symbols are the experimental results and curves that fit into Equation (4) with n = m = 3. In panels from (**c**–**e**), the composition dependences of the surface tension for the different studied systems are displayed. The insets identify the mixtures and the temperatures of the measurements, 298.15 K, unless the other value is specified. The arrows indicate the positions of the pseudo plateaus.

**Figure 2 ijms-24-02276-f002:**
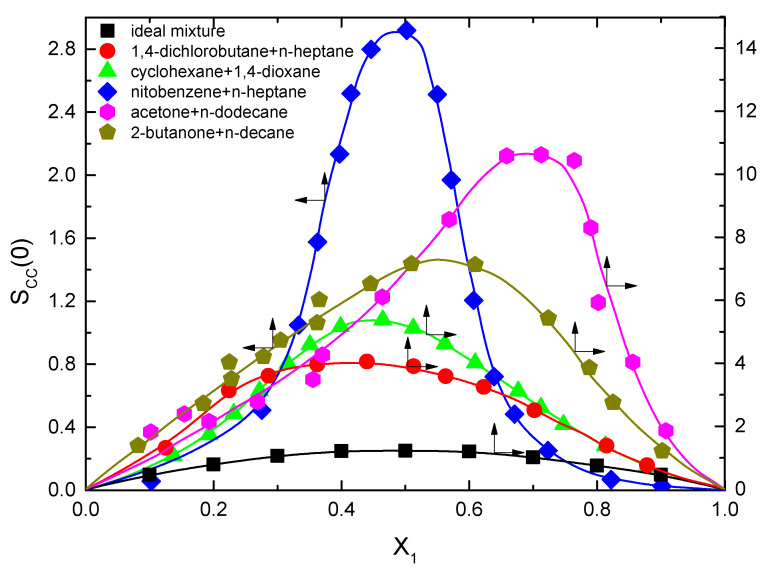
Composition dependence of the concentration-concentration correlation function at 298.15 K. The dashed-dotted line corresponds to an ideal mixture. Notice that some of the curves refer to the left ordinate axis, while the others refer to the right axis. Some of the data shown have been reproduced with permission from our previous publication [5].

**Figure 3 ijms-24-02276-f003:**
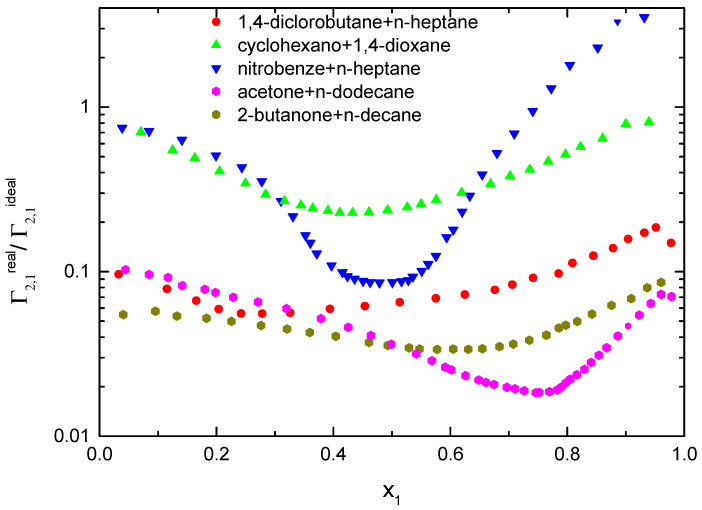
Ratio of the relative surface concentration, Γ_2,1_, for real and ideal mixtures.

## Data Availability

Data are available upon request.

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
