# Peer review of "Anomalous Concentration Dependence of Surface Tension and Concentration-Concentration Correlation Functions of Binary Non-Electrolyte Solutions"

_ijms, 2023, doi:10.3390/ijms24032276_

Round 1

Reviewer 1 Report

The manuscript by C. Carbone et al. contains very interesting results on the surface tension of a few binary mixtures of non-electrolytes with plateau regions of the surface pressure isotherms. The combination of these data with the results on depolarized light scattering allowed the authors to conclude that the calculation of the adsorbed amount according to the Gibbs adsorption equation, in which the activities of two components are substituted by mole fractions, is not acceptable in a broad range of mole fractions. This subject can be interesting for a broad readership of IJMS. Unfortunately, the manuscript is difficult for reading because it contains a significant number of misprints and some statements require additional proofs and explanations. The main concern of the reviewer is the connection of the surface tension plateau with the concentration fluctuations. Other explanations are also possible. Do the strong fluctuations disappear outside the plateau regions? It is difficult to expect this behavior. It is more probable that the plateau regions are caused by a heterogeneity of the liquid surface (surface phase transitions) when the changes of the composition result only in the changes of the areas occupied by different surface phases at a constant surface tension. The recommendation is to discuss this possibility in the manuscript. The manuscript can be recommended for publication only after some improvements as indicated below.

L. 44. Please explain why the given derivative coincides with the osmotic compressibility and define μi.

L. 51. The relation for the chemical potential is not correct.

L. 65.  The definition of Г2,1 is not correct. According to Gibbs, this is the adsorbed amount of the component 2 relative to the component 1.

Ls. 68-71. Relation (2) is usually applied for extremely dilute solutions. Do the authors know such applications at molar fractions close to 0.5 as in the manuscript?

Ls. 90-91. What are the “excess consequences”?  

Ls. 103-105. What theoretical models are under consideration?

Ls. 110, 116, 119. What is the excess surface tension? Please define.

Ls. 127-131. In the plateau regions (∂γ/∂x2)T,P  is almost zero. How one can determine reliable values of Г2,1? What is the precision of Scc(0)?

L. 133. “the maxima of the curves are up two orders of magnitude higher”. Is it correct? Fig. 2 shows that the maxima are about one decimal order of magnitude higher than the results for the ideal behavior.

L. 137. Where is “dashed-dotted line”?

L. 147. What means “the excess surface adsorption”?

Ls. 149-151. “…the high values of the concentration-concentration correlation function indicate strong concentration fluctuations, which are associated to strong concentration fluctuations…” What does this mean?

Ls. 151-153. Can the authors give some examples of ideal binary mixtures at mole fractions of about 0.5?

Ls. 157-159. It is well known that the purity of the substances indicated by a supplier is not important for surface studies because slight and practically undetectable traces of impurities can concentrate in the surface layer and change the surface tension. In this case only surface purification can give more or less reliable results. Did the authors use a simple test of the purity with a Langmuir trough and a moving barrier? What is the surface age of the results in Fig. 1? The kinetic dependencies of the surface tension are quite important to estimate roughly the purity of the substances. The recommendation is to show the corresponding kinetic dependencies, maybe in the supplementary information.  

Ls. 160-163. All the conclusions of the manuscript are based on very precise measurements of the surface tension. Meantime, the applied experimental methods have different sources of errors and they are not discussed in the manuscript. For example, the results of the ring methods depend on the ring motion in the course of measurements. The results of Wilhelmy plate and pendant drop methods depend on the proximity of the boundary angle to zero. All the methods require some time for the equilibration. What is this time?

Please explain which experimental points in Fig. 1 were obtained by the ring, plate of pendant drop.

Ls. 186-189. Direct estimations of the surface layer thickness, for example by the ellipsometry, would be useful.

Eq. (A.1). The second equality is incorrect. The expression before the arrow is a repetition.

Eq. (A.4). Please define all the terms.

L. 225. Relation (A.4) is not a definition of Rc

Ls. 240-241. “According to Segudovic and Dezelic, R* can be expressed, in a first approximation, as…” Please give a reference.

Ls. 247-248. Please give a reference.

Author Response

  1. 44. Please explain why the given derivative coincides with the osmotic compressibility and define μi.

mi has now been defined as the chemical potential of component i.

The Reviewer is right. There is a typo in the definition of the osmotic compressibility, we have written g instead of Xi. We have corrected it in the revised version.

  1. 51. The relation for the chemical potential is not correct.

We have corrected the error in the new version

  1. 65.  The definition of Г2,1 is not correct. According to Gibbs, this is the adsorbed amount of the component 2 relative to the component 1.

The definition of  has been modified according to the suggestion.

Ls. 68-71. Relation (2) is usually applied for extremely dilute solutions. Do the authors know such applications at molar fractions close to 0.5 as in the manuscript?

Strictly speaking, the use of equation (2) is not limited to low concentration as far as the chemical potential is used in the derivative instead of the concentration. In the way that it is usually applied, the Reviewer is right because assuming that the activity is equal to the concentration is valid only for ideal or very highly diluted solutions. This is not true for intermediate or high concentrations, and it is not clear whether for ionic surfactant or polymer solutions it can be used except for extremely diluted concentrations, e.g., see the theories developed by V. Feinerman and R. Miller and coworkers. Taking this into account, it is right why we have used the polarized light scattering results to avoid assuming any solution model.

The use of the ideal solution approximation in Figure 3 is not because we assume it valid, but simply because we believe that it allows the reader to grasp faster the huge difference with respect to the ideal solution assumption. We have clarified this point in the revised version of the manuscript.

Ls. 90-91. What are the “excess consequences”?  

Mixtures of organic components are often used in industry as solvents of polymers or to form layers onto immiscible liquids such as water for spreading hydrophobic insecticides. Having an air/liquid surface much enriched in one of the solvents can modify the adsorption of those molecules at the surface because of the different solvent quality of each of them.

Ls. 103-105. What theoretical models are under consideration?

The relationships between the osmotic compressibility and the concentration-concentration correlation functions are based on the integral theory of fluids derived by J.C. Kirkwood and F. Buff, J. Chem. Phys. 19, 774 (1951). In the case of binary mixtures, the structure has been better discussed in terms of the partial structure factors, aij, where is the wave vector, and i and j refer to the two types of molecules [Y. Irai and Y. Arai, Fluid Phase Equilib. 9, 201 (1982)]. The partial structure factors are defined as

Where gij(r) is the radial distribution function of molecules i and j, is the vector joining the center of the two molecules, and r the density. In the exponential, .

The long-wave limit of the partial structure factors, q®¥, are directly related to the so-called Kirkwood-Buff integrals [J.C. Kirkwood and F. Buff, J. Chem. Phys. 19, 774 (1951); D.A. McQuarrie, “Statistical Mechanics”, Harper & Row, New York, 973]

that are related to the osmotic compressibility [C.G. Gray and K.E. Gubbins, “Theory of Molecular Fluids”, Oxford U.P., London, 1984] through

Vi being the molar volume of component I, V the molar volume of the mixture, kT the isothermal compressibility, and D is given by

Ls. 110, 116, 119. What is the excess surface tension? Please define.

The reviewer is right. The usual definition of excess functions do not strictly apply to the surface tension. We tried to simplify the nomenclature and have really made a mistake, that has also been pointed out by Reviewer 1. In the new version we just define the function

without making reference to any excess function idea for avoiding any confusion.

Ls. 127-131. In the plateau regions (∂γ/∂x2)T,P  is almost zero. How one can determine reliable values of Г2,1? What is the precision of Scc(0)?

The precision of Scc(0) is about 3%, although it depends on the difference of the refractive index of the two solvents and of their polarizability tensor.

Our surface tension results do not show any maximum or minimum, but only a region in which the slope, , decreases at intermediate mole fractions, and then increases again, without being zero at any point. However, we have found interesting to mention the for aneotropic points for comparison with the results obtained in this work.

  1. 133. “the maxima of the curves are up two orders of magnitude higher”. Is it correct? Fig. 2 shows that the maxima are about one decimal order of magnitude higher than the results for the ideal behavior.

The Reviewer is right, the maximum difference is 14.3 vs. 0.25. We have corrected this error in the revised version.

  1. 137. Where is “dashed-dotted line”?

The Reviewer is right, we refer to the black line. We have corrected this mistake in the revised version.

  1. 147. What means “the excess surface adsorption”?

We should have defined as the “adsorbed amount of the component 2 relative to the component 1”. This has been modified along the text.

Ls. 149-151. “…the high values of the concentration-concentration correlation function indicate strong concentration fluctuations, which are associated to strong concentration fluctuations…” What does this mean?

“Strong concentration fluctuations” means that the local concentration around a molecule of type two, X2,local, is quite difference from the average one, X2. However, X2,local around one molecule of type 2 is not exactly the same than around another molecule of the same type at a different position at the same time. Of course, the average of X2,local for all the molecules of type 2 have to be equal to the average value of X2. This explanation has been added in the new version.

Ls. 151-153. Can the authors give some examples of ideal binary mixtures at mole fractions of about 0.5?

For an ideal mixture Scc(0) = X1·X2, thus the maximum corresponds to Scc(0) =0.25 for X2 = 0.5. Probably this point arises from our mistake in the legend of Figure 2 that the Reviewer pointed out in his/her comment about L. 133

Ls. 157-159. It is well known that the purity of the substances indicated by a supplier is not important for surface studies because slight and practically undetectable traces of impurities can concentrate in the surface layer and change the surface tension. In this case only surface purification can give more or less reliable results. Did the authors use a simple test of the purity with a Langmuir trough and a moving barrier? What is the surface age of the results in Fig. 1? The kinetic dependencies of the surface tension are quite important to estimate roughly the purity of the substances. The recommendation is to show the corresponding kinetic dependencies, maybe in the supplementary information.  

We agree with the Reviewer that the interfacial properties are very sensitive to the existence of impurities. It is well known that in the case of surfactant solutions, e.g., sodium dodecyl sulfate, the existence of hydrophobic impurities strongly perturbs the concentration dependence of the surface tension close to the C.M.C. In the case of the solvents used in this work, their purity is higher than for most of the surfactants. The less pure compound is chloronaphtalene (98.8 wt%). We have recrystallized after the Reviewer comments, and the surface tension results have not changed. The main difference with respect to surfactant solutions is that both compounds used here are organic, i.e., very hydrophobic substances, whereas in the case of surfactants one uses water as solvent in most cases. Therefore, it seems reasonable that tiny amounts of hydrophobic impurities will have much less effect in the systems studied in this work.

Following the Reviewer’ suggestion we have performed adsorption kinetic measurements for three mole fractions, 0.2 and 0.5 and 0.8, for 2-butanone+n-decane, and chloronaphtalene+2,2,4-trimethylpentane. The equilibration time in the plate and in the ring tensiometers were less than two minutes when each of the components had been thermostated prior to mixing. This is in accordance to the low viscosity and high diffusion coefficients of the solvents. Our techniques do not allow us to follow a detailed enough kinetic measurements for such short stabilization times.

Moving barrier measurements using a Langmuir balance inducing a 400% reduction of the initial area. No change of the surface tension was detected within the experimental uncertainty.

This has been discussed in the revised version of the manuscript.

Ls. 160-163. All the conclusions of the manuscript are based on very precise measurements of the surface tension. Meantime, the applied experimental methods have different sources of errors and they are not discussed in the manuscript. For example, the results of the ring methods depend on the ring motion in the course of measurements. The results of Wilhelmy plate and pendant drop methods depend on the proximity of the boundary angle to zero. All the methods require some time for the equilibration. What is this time?

The comment of the Reviewer is correct. Each of the techniques used has different sources of error. This is why we decided to perform the measurements using three different experimental techniques, thus being sure that the “anomalous” concentration dependence was not an experimental artefact. As it can be observed in Figure 1, the agreement is very good, always within the combined experimental uncertainty (+/- 0.1 mN/m)

Please explain which experimental points in Fig. 1 were obtained by the ring, plate of pendant drop.

Measurements with ring, plate and pendant drop were used randomly within the whole concentration range, and in some cases measurements with the three techniques were performed for the same compositions for ensuring the validity of our discussion.

Ls. 186-189. Direct estimations of the surface layer thickness, for example by the ellipsometry, would be useful.

Indeed an estimation of the surface thickness would be interesting in order to experimentally justify the tentative explanation given for the values of G2,1, however the small difference of the refractive indices of the solvents has prevented to obtain reliable values for the thickness. The use of neutron reflectivity would lead good results because either of the solvents could be deuterated, thus allowing one to obtain several independent reflectivity curves for each mixture.

Eq. (A.1). The second equality is incorrect. The expression before the arrow is a repetition.

The Reviewer is right. The equation has been corrected.

Eq. (A.4). Please define all the terms.

We have done it in the revised version.

  1. 225. Relation (A.4) is not a definition of Rc

We have rewritten the equation y the correct form.

Ls. 240-241. “According to Segudovic and Dezelic, R* can be expressed, in a first approximation, as…” Please give a reference.

We have added the reference

Ls. 247-248. Please give a reference.

We have given the reference

Thank you very much for your helpful comments and suggestions.

Reviewer 2 Report

The main aim of this work is to explain the stationary points observed in the interfacial tension–concentration at isothermal conditions for miscible binary mixtures of non-electrolytes fluids. In order to describe the observed plateau, the authors use the “excess surface tension” to visualize the quoted behavior through the W-Shape behaviour. Then they combine the concentration fluctuations to the relative Gibbs adsorption, where concentration fluctuations provide a route to explain the observed plateau.

According to the reported information, the results provide a route to validate the approximation of the ideal solution. However, some important points are not detected or explained as they are described below:

1. According to the definition of the excess function, surface tension is not an excess property.

2. At a stationary point in the interfacial tension – concentration at isothermal conditions or aneotropy point, the relative Gibbs adsorption is equal to zero (see Eq. 6)

3. Based on the work of McLure, the behavior of relative Gibbs adsorption is well-understated at the aneotropy point. It is advised to reconsider the proposed discussion and also revise the following reference J. Chem. Thermodynamics 43 (2011) 1395–1400 and references therein.

4. The manuscript's organization needs to be redefined: 2. Materials and Methods, and 3. Results and Discussion

Author Response

  1. According to the definition of the excess function, surface tension is not an excess property.

The reviewer is right. The usual definition of excess functions do not strictly apply to the surface tension. We tried to simplify the nomenclature and have really made a mistake, that has also been pointed out by Reviewer 1. In the new version we just define the function

without making reference to the excess function idea for avoiding any confusion.

  1. At a stationary point in the interfacial tension – concentration at isothermal conditions or aneotropy point, the relative Gibbs adsorption is equal to zero (see Eq. 6)

This is true, however the experimental surface tension results do not show any maximum, minimum or inflection point, but only a marked change of slope that is not zero in any concentration value. This is why the excess surface concentration is not zero.

  1. Based on the work of McLure, the behavior of relative Gibbs adsorption is well-understated at the aneotropy point. It is advised to reconsider the proposed discussion and also revise the following reference J. Chem. Thermodynamics 43 (2011) 1395–1400 and references therein.

As stated above, our surface tension results do not show any maximum or minimum, but only a region in which the slope, , decreases at intermediate mole fractions, and then increases again, without being zero at any point. However, we have found interesting to mention the work of McLure for aneotropic points, as a difference to the results obtained here.

  1. The manuscript's organization needs to be redefined: 2. Materials and Methods, and 3. Results and Discussion

This a helpful suggestion. We have reorganized the manuscript.

Thank you very much for your helpful comments and suggestions

Round 2

Reviewer 1 Report

The authors corrected their manuscript, answered all the questions of the reviewer and the new version of the manuscript can be published as it is.

Reviewer 2 Report

In the revised form, the authors are carried out all my queries.